# Targeting of nanoparticles to the cerebral vasculature after traumatic brain injury

**Serena Omo-Lamai**[1☯], **Jia Nong**[2☯], **Krupa Savalia**[3☯], **Brian J. Kelley**[4☯], **Jichuan Wu**[5], **Sahily Esteves-Reyes**[6], **Liam S. Chase**[5], **Vladimir R. Muzykantov**[2], **Oscar A. Marcos-Contreras**[2], **Jean-Pierre Dollé**[4], **Douglas H. Smith**[4‡*], **Jacob S. Brenner**[2,5‡*]

1 Department of Bioengineering, School of Engineering and Applied Sciences, University of Pennsylvania, Philadelphia, Pennsylvania, United States of America, 2 Department of Systems Pharmacology and Translational Therapeutics, Perelman School of Medicine, University of Pennsylvania, Philadelphia, Pennsylvania, United States of America, 3 Departments of Neurology & Neurological Surgery, University of California—Davis, Sacramento, California, United States of America, 4 Department of Neurosurgery, Perelman School of Medicine, University of Pennsylvania, Philadelphia, Pennsylvania, United States of America, 5 Department of Medicine, Division of Pulmonary Allergy and Critical Care, Perelman School of Medicine, University of Pennsylvania, Philadelphia, Pennsylvania, United States of America, 6 Department of Neurology, Perelman School of Medicine, University of Pennsylvania, Philadelphia, Pennsylvania, United States of America

☯ These authors contributed equally to this work.
‡ DHS and JSB authors contributed equally
* jacob.brenner@pennmedicine.upenn.edu (JSB); smithdou@pennmedicine.upenn.edu (DHS)

**Data Availability Statement:** All relevant data are within the manuscript and its Supporting Information files.

**Funding:** S.O. received funding from the American Heart Association (Grant 23PRE1014444). J.N.

## Abstract

Traumatic brain injury has faced numerous challenges in drug development, primarily due to the difficulty of effectively delivering drugs to the brain. However, there is a potential solution in targeted drug delivery methods involving antibody-drug conjugates or nanocarriers conjugated with targeting antibodies. Following a TBI, the blood-brain barrier (BBB) becomes permeable, which can last for years and allow the leakage of harmful plasma proteins. Consequently, an appealing approach for TBI treatment involves using drug delivery systems that utilize targeting antibodies and nanocarriers to help restore BBB integrity. In our investigation of this strategy, we examined the efficacy of free antibodies and nanocarriers targeting a specific endothelial surface marker called vascular cell adhesion molecule-1 (VCAM-1), which is known to be upregulated during inflammation. In a mouse model of TBI utilizing central fluid percussion injury, free VCAM-1 antibody did not demonstrate superior targeting when comparing sham vs. TBI brain. However, the administration of VCAM-1-targeted nanocarriers (liposomes) exhibited a 10-fold higher targeting specificity in TBI brain than in sham control. Flow cytometry and confocal microscopy analysis confirmed that VCAM-1 liposomes were primarily taken up by brain endothelial cells post-TBI. Consequently, VCAM-1 liposomes represent a promising platform for the targeted delivery of therapeutics to the brain following traumatic brain injury.

received funding from the American Heart Association (Grant 916172). B.J.K received funding from National Health Institute (K08-NS110929). O. A.M.-C. received funding from the American Heart Association (Grant 19CDA34590001). J.S.B. and V. M.R. received support from the Cardiovascular Institute of the University of Pennsylvania. V.M.R. received funding from National Institute of Health (R01 HL155106, R01 HL128398, R01 HL143806). J.S.B. received funding from National Institute of Health (K08-HL-138269, R01-HL-153510, R01-HL-160694, R01-HL-157189, R21-AI-166778-01). D.H.S received funding from the Paul G. Allen Family Foundation and National Institute of Health (U54 NS115322). AHA: https://www.heart.org/en/get-involved/ways-to-give?form=FUNPHPZDXBX&s_src=210L511AEMG&s_subsrc=fy23_jun_sem_google_text_&utm_medium=paid&utm_campaign=dr+fy23+june&utm_source=sem+google&utm_content=prospecting-remarketing+sem+general&utm_term=text&gad=1&gclid=CjwKCAjwp6CkBhB_EiwAlQVyxUMZ_FbtTs7VO9Lig6RdFLrTCBLy_fDCRjQoyJToLTNRKw3B3axoQhoC4ZwQAvD_BwE&gclsrc=aw.ds CVI: https://www.med.upenn.edu/cvi/funded-dream-teams.html NIH: https://www.nih.gov/grants-funding The funders had no role in study design, data collection and analysis, decision to publish, or preparation of the manuscript.

Competing interests: The authors have declared that no competing interests exist.

## Introduction

Traumatic brain injury (TBI) results in 230,000 hospitalizations and 50,000 deaths in the US each year annually [1]. While there have been multiple TBI clinical treatment trials, to-date none have been successful [2–9]. One major challenge is that most candidate drugs have poor accumulation in the brain, due to exclusion by the blood-brain barrier (BBB). One potential solution to this problem is to develop targeted drug delivery vehicles that can localize drugs to the injured brain.

Two major platforms currently exist for targeted drug delivery. First, monoclonal antibodies that can be conjugated to small molecule drugs or siRNA, forming antibody-drug conjugates (ADCs) [10–14]. Second, nano-scale drug carriers (nanocarriers) that can be loaded with drugs and then covalently conjugated to antibodies that target a particular organ or cell type [15–17]. Both of these strategies have been investigated for general brain delivery [18,19] but with little attention to TBI specifically. Therefore, here we examined the potential of brain targeting after TBI through monoclonal antibodies and targeted nanocarriers.

In contrast to previous strategies of brain delivery that focused on delivery drugs *beyond* the BBB to the brain parenchyma (18, 20–22), we aim to target antibodies and nanocarriers specifically *to* the BBB itself because of the following reasons. Due to the large accessible surface area of BBB endothelium after intravenous (IV) injection, by delivering cargo drugs *to* the endothelial cells of the BBB, we have achieved the highest reported delivery to the brain of any IV-based drug therapy, i.e. endothelial cell adhesion molecule (CAM)-targeted nanocarrier delivery of >10% injected dose to the brain (via intra-arterial red blood cell-hitchhiking) [20,21]. Thus, targeting antibodies and nanocarriers to the endothelial cells of the BBB offers a way to concentrate cargo drugs in the brain.

Following a TBI, the BBB is significantly disrupted with altered permeability being detected multiple years later [22–24]. TBI-induced BBB dysfunction has been attributed to an increase in paracellular transport through the loss of tight junction proteins, and an increase in larger molecules and proteins through transcytosis [25,26]. Toxic plasma proteins (e.g., complement C3 and thrombin) that can promote vasogenic edema, bind with protease active receptors and induce neuroinflammation. This acute BBB disruption may result in worse long-term outcome after TBI [27]. Thus, targeted drugs that can mitigate BBB permeability might ameliorate key secondary components of TBI [28]. Therefore, a major goal of TBI therapeutics should be the closure of the leaky BBB, which may be achieved through targeting drugs to the brain's endothelium.

To target the brain endothelium, we have previously shown that antibodies and nanocarriers that bind endothelial CAMs achieve very high brain uptake. In this proof of principle study, we tested vascular CAM-1 (VCAM-1), which is upregulated in endothelial cells during inflammation, and has shown significant brain delivery in multiple brain disorders and in other diseases [21,29–34]. While our goal was to determine whether these targeting antibodies are useful for drug-targeting in TBI, we initially aimed to answer the following scientific questions: First, since TBI has significant capillary leak, would untargeted antibodies (control immunoglobulin G, IgG) simply leak into the brain, and thereby achieve better brain uptake than VCAM-1-antibodies? Second, does TBI change biodistribution within the brain and body of VCAM-1-targeted antibodies or nanocarriers? And finally, do antibodies and targeted nanocarriers behave comparably in the setting of TBI?

## Materials and methods

### Materials

DPPC (dipalmitoyl phosphatidylcholine), cholesterol, and DSPE-PEG2000-azide (1,2-distearoyl-sn-glycero-3-phosphoethanolamine-N-[azido(polyethylene glycol)-2000] (ammonium

salt)) were purchased from Avanti Polar Lipids (Alabaster, Alabama). All other chemicals and reagents were purchased from SigmaAldrich (St. Louis, MO), unless specifically noted.

## Liposome preparation and characterization

Liposomes were formulated using the thin-film hydration method. Lipids were dissolved in chloroform and combined in a borosilicate glass tube. Chloroform was evaporated by blowing nitrogen over the solution until visibly dry (approximately15 minutes) then putting the tube under vacuum for greater than 1 hour. Dried lipid films were hydrated with phosphate buffered saline to a total lipid concentration of 20mM. The rehydrated lipid solution was vortexed and sonicated in a bath sonicator until visually homogeneous (approximately 1 minute each of vortexing and sonication). The solution was then extruded twenty-one times through a 0.2 μm polycarbonate filter. Liposomes were heated to approximately 50°C (just above the phase transition temperature of DPPC) during vortexing and extrusion. Dynamic light scattering (DLS) measurements of hydrodynamic particle size, distribution, and polydispersity index were made using a Zetasizer Pro ZS (Malvern Panalytical, Malvern UK).

## Antibody modification

To conjugate to immunoliposomes, antibodies were functionalized with DBCO by reacting with a 5-fold molar excess of DBCO-PEG4-NHS ester for 30 minutes at room temperature. The unreactive compound was removed with centrifugation using a molecular weight cutoff filter or G-25 Sephadex Quick Spin Protein column (Roche Applied Science, Indianapolis, IN).

For biodistribution studies, monoclonal antibodies were radiolabeled with Na$^{125}$I using Pierce Iodogen radiolabeling method [21]. Briefly, tubes were coated with 100 μg of Iodogen reagent. The antibody (1–2 mg/mL) and Na$^{125}$I (0.25 μCi/μg protein) were placed on ice for 5 minutes. The excessive materials were purified using Zeba desalting spin columns (Thermo-Fisher Scientific).

## Liposome conjugation

Liposome conjugation to antibodies was carried out using DBCO-azide copper-free "click chemistry" Azide functionalized liposomes were incubated overnight with DBCO-modified antibodies at 37°C. Immunoliposomes (∼50 mAbs/liposome) were purified from residual antibodies using size-exclusion chromatography (Sepharose 4B-Cl; GE Healthcare, Pittsburg PA). For liposome biodistribution studies, radiolabeled untargeted IgG (~10% of total antibody) was doped into liposome conjugation for radio tracing.

## TBI animal model

All animal experiments were approved by the Institutional Animal Care and Use Committee of The University of Pennsylvania. Adult male C57BL/6 mice aged 10–15 weeks (26.6 +/- 1.7 g; The Jackson Laboratory) were housed in an animal facility with a 12-hour light-dark cycle and fed standard chow and water ad libitum. Mice were subjected to either a sham injury, as described below, or a central fluid percussion injury (cFPI) which was modified from those previously described [35–37]. Each animal was anesthetized in a chamber with 4% isoflurane in 100% oxygen. After induction, the scalp was prepared with betadine for sterilization and placed in a stereotactic frame that was fitted with a nose cone to maintain anesthesia with 1–2% isoflurane in 100% oxygen. A midline sagittal incision was made to expose the skull from bregma to lambda. The skull was cleaned and dried prior to performing a 3.0mm circular

craniotomy by positioning a trephine along the sagittal suture midway between bregma and lambda. A sterile Leur-lock syringe hub was cut away from a 20-gauge needle and then affixed to the craniotomy site using cyanoacrylate. Following confirmation of seal integrity between the hub and the skull, dental acrylic was applied around the hub to provide further stability during injury induction as well as in the sham animals. The dental acrylic was allowed to harden and topical bacitracin and lidocaine ointments were applied to the incision site. Animals were removed from the anesthesia and monitored in a warmed cage to allow for full recovery prior to injury induction.

Each animal was again anesthetized the same day with 4% isoflurane in 100% oxygen. The Leur-lock syringe hub was filled with normal saline prior to connecting the animal to the fluid percussion apparatus while laying in the right lateral decubitus position. An injury of mild-to-moderate severity (average atm +/- STD: 1.745 +/- 0.129) was administered by releasing the pendulum onto a fluid-filled piston in order to induce a brief fluid pressure pulse upon the intact dura. The pressure pulse was measured by a transducer, which was captured by the computer software, with recording of the peak pressure (average psi +/- STD: 25.431 +/- 1.867). Animals in the control group (sham) did not experience a fluid pressure pulse but were attached to the fluid percussion apparatus to mimic a sham injury. The hub with dental acrylic was then removed from the skull en bloc; bleeding (if present) was controlled with Gelfoam and the midline sagittal incision was rapidly sutured before recovery from anesthesia. Topical bacitracin and lidocaine ointments were applied to the sutured scalp incision and the animals were visually monitored for recovery of spontaneous respiration. The duration of transient unconsciousness from anesthesia and/or injury was determined by measuring the length of time for each animal to recover the righting reflex. After recovery of the righting reflex, animals were placed in a warmed cage to ensure maintenance of normothermia and to allow for full recovery of consciousness.

## Flow cytometry in brain tissue

Brains were dissociated as previously described (25, 30). Briefly, brains were first manually disaggregated by repeated pushing of tissue through needles with varying gauges. The resulting suspension was filtered through a 100 μm nylon strainer, centrifuged, and resuspended in dispase (2.5 U/mL, ThermoFisher) for 1 hour. The filtered resuspension was then passed through a 70 μm filter and treated with 600 units/mL of DNase I (grade I, Sigma Aldrich) prior to centrifugation and demyelination of the cell pellet using a standard isotonic Percoll (SIP) gradient. The resulting demyelinated pellet was suspended in ACK lysis buffer (Quality Biological) for red blood cell lysis prior to staining with fluorescent antibodies. The following markers were used to assess cell-type distribution: endothelial cells (CD31-high, CD45-neg), leukocytes (CD45-high) and microglia (CD45-mid). Flow cytometry was performed using an Accuri C6 cytometer (BD).

## Confocal imaging

Twenty-four hours-post injury, TBI mice were injected with fluorescently labeled VCAM-1-targeted liposomes. Twenty-five minutes later, Alexa fluor- 647 labeled Mec13.3 (Biolegend) was injected and circulated for 5 minutes before the animals were sacrificed. After perfusion with cold PBS, brains were harvested and freshly frozen. 10 um brain slices were cryosectioned for imaging using Leica TCS SP8 confocal microscopy.

## Biodistribution

TBI or sham mice were injected intravenously with iodinated monoclonal antibodies (5μg/animal) or radiolabeled immunoliposomes (10mg/kg lipid, ~6e11 liposomes/animal) 24 hours

following sham injury or TBI. Animals were sacrificed by exsanguination under anesthesia 30 minutes after injection and perfused with 20mL of PBS prior to organ procurement. The amount of radioactivity in blood and organs was measured using a gamma counter (Wizard2, PerkinElmer, Waltham, MA).

### Statistics

All results are expressed as mean ± SEM. Statistical analyses were performed using GraphPad Prism 8 (GraphPad Software, San Diego, CA). * denotes p<0.05, ** denotes p<0.01, *** denotes p<0.001, **** denotes p<0.0001.

## Results

### VCAM-1 antibody localizes significantly to the brain in sham and TBI mice

To assess the accessibility of VCAM-1 antibody in TBI brain, twenty-four hours following a sham or mild-to-moderate central fluid percussion injury to mice (**Fig 1A**) [37], we IV injected radiolabeled monoclonal antibody (5μg mAb per animal), or untargeted control IgG. The antibodies were allowed to circulate for 30 minutes prior to animal sacrifice (**Fig 1B**). VCAM-1 antibody accumulated in the brain at significantly higher levels than control IgG antibodies in both sham and TBI mice (**Fig 1C and 1D**, **S1 Table**). In sham mice, VCAM-1 antibody was

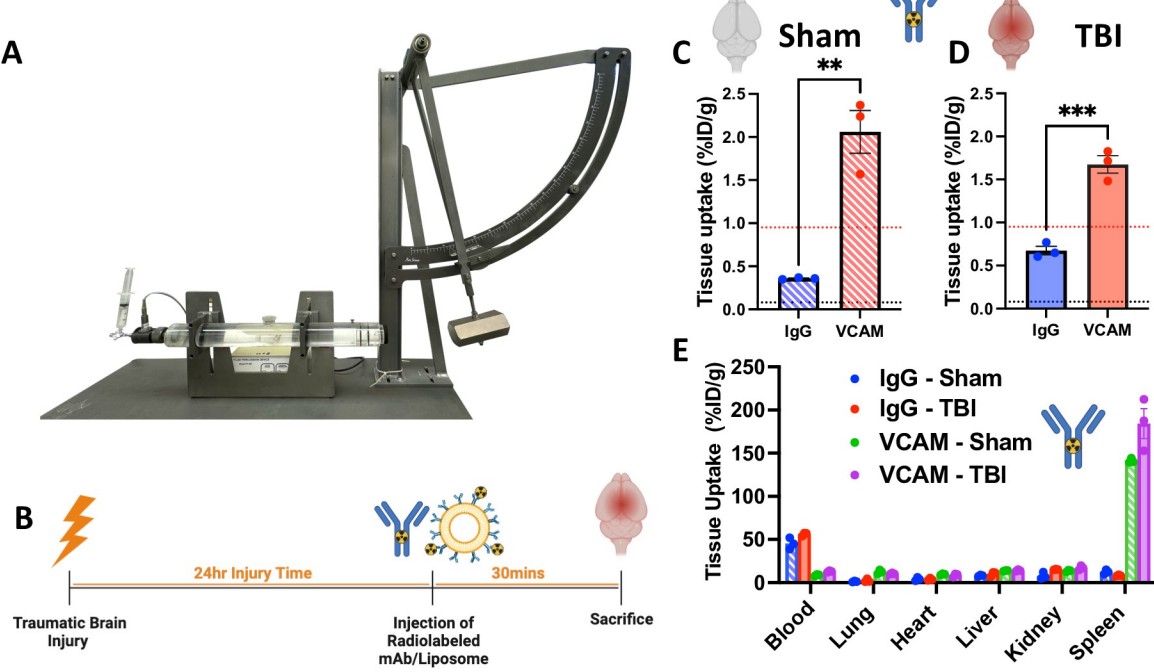

**Fig 1. In TBI mice, VCAM-1 antibody has significantly higher brain uptake than untargeted IgG control.** (A) To prepare mice for the central fluid percussion injury, a sterile Leur-lock syringe hub was first attached to the craniotomy site which was then filled with saline. Using a fluid percussion injury device, a mild-to-moderate injury was administered by releasing the pendulum onto a fluid-filled piston in order to exert fluid pressure upon the dura. (B) Timeline of biodistribution, flow cytometry and histology experiments of monoclonal antibodies or targeted-nanocarriers against VCAM-1 or untargeted IgG control. Comparing the brain uptake of VCAM-1 to IgG in sham (C) and TBI (D) shows that antibodies against endothelial targets accumulate in the brain significantly more than control IgG antibodies (%ID/g: % of injected dose per gram of tissue). Notably, there is no significant difference between the brain uptake in sham and TBI mice. Dashed line represents naïve level of IgG (black) vs. VCAM-1 (red). (E) Similarly, no significant differences were observed in the biodistribution of VCAM-1 and IgG antibodies in other organs between sham and TBI mice. N≥3 and data shown represents mean ± SEM; Comparisons were made by student's t-test. *p<0.05, ***p<0.001.

taken up ~6-fold more than untargeted IgG control. By contrast, in TBI mice, VCAM-1 antibody was taken up ~2.5 fold more than IgG. It is worth noting that the IgG accumulation in the TBI brain is ~8 fold higher than that of naïve brain (black dashed line). Notably, IgG circulates well and accumulates minimally in the healthy brain. These results suggested that TBI induces capillary leakage, though at this point we cannot say if IgG extravasation into the TBI brain is due to transport that is transcellular, paracellular, or via frank hemorrhage. Additionally, the extravasated IgG in TBI brain contributed to the lower fold-improvements of VCAM-1 antibody vs. IgG control in TBI vs. sham brain.

The whole body biodistribution of VCAM-1 and IgG antibodies was nearly identical between the sham and TBI groups, with VCAM-1 antibody showing highest uptake in the spleen (**Fig 1E**, **S1 Table**). Thus, TBI does not affect non-brain organs' uptake of VCAM-1 (noting that these change in other acute injuries [38]). The notable differences in non-brain biodistribution between VCAM-1 and IgG are all ones that we have reported in naive animals before [21]: IgG has high blood levels (without a target to bind, it circulates for days), while VCAM-1 has high spleen levels.

Thus, in TBI, antibody targeting VCAM-1 accumulate in the brain at higher levels than untargeted control IgG, and TBI does not change their biodistribution outside the brain.

## VCAM-1 liposomes efficiently target the brain in TBI mice

Having quantified the uptake and circulation of VCAM-1 antibody, we next wanted to evaluate their performances when conjugated onto liposomes, which are present in FDA approved formulations [39]. Liposomes were prepared using the thin film hydration method and to their surface we covalently conjugated radiolabeled monoclonal antibody that bind to VCAM-1 or control (untargeted) IgG. Antibody-conjugated liposomes were then IV-injected into sham or TBI mice for a circulation period of 30 minutes.

In sham mice, there was no significant difference in the brain uptake of VCAM-1-targeted liposomes compared to control IgG liposomes (**Fig 2A**, **S2 Table**). However, in TBI mice, VCAM-1-liposomes had a 10-fold higher brain uptake than IgG liposomes. In addition, TBI led to significantly higher brain delivery of VCAM-1-targeted liposomes, when compared to sham. In the whole body biodistributions, VCAM-1-liposomes performed similarly to their free antibody counterparts, with VCAM-1 showing significant localization to the spleen, and there is a higher liver uptake with targeted nanoparticles than free antibodies (**Fig 2B**, **S2 Table**). Since spleen is the major reservoir of VCAM-1 (shown in **Fig 1E**) and clearance organ of nanocarriers, we wanted to evaluate the immunospecificity (IS) of VCAM-1 targeted liposomes in spleen and brain, by accounting for the effect of blood driven delivery and non-specific delivery of IgG control. Using the IS index calculation: (%ID/g VCAM-1-organ/%ID/g VCAM-1-blood)/(%ID/g IgG-organ/%ID/g IgG-blood), we found that in sham animals, the IS index of brain and spleen are similar ($2.539 \pm 0.431$ vs. $2.251 \pm 0.082$, $p = 0.055$). However, in TBI animals, the IS index of brain in significantly higher than spleen ($14.805 \pm 1.273$ vs. $9.735 \pm 0.657$, $p = 0.024$). It suggests that VCAM-1-targeted liposomes have specific targeting to the TBI-injured brain.

Intriguingly, when we compare the brain IS index of VCAM-1-antibodies vs VCAM-1-liposomes, we find a stark difference in how they respond to TBI. The accumulation of VCAM-1 free antibody in the brain is unaffected by TBI, actually inferior to sham control. Due to the leaky blood-brain barrier after TBI, IgG uptake in the TBI brain is 1.86-fold higher than in sham brain. When antibody uptake in other tissues are comparable (e.g. the blood % ID/g of IgG and VCAM-1 antibody are within the range of what we've been observed in mice), the small change in brain uptake will significantly impact on the calculated outcome of the

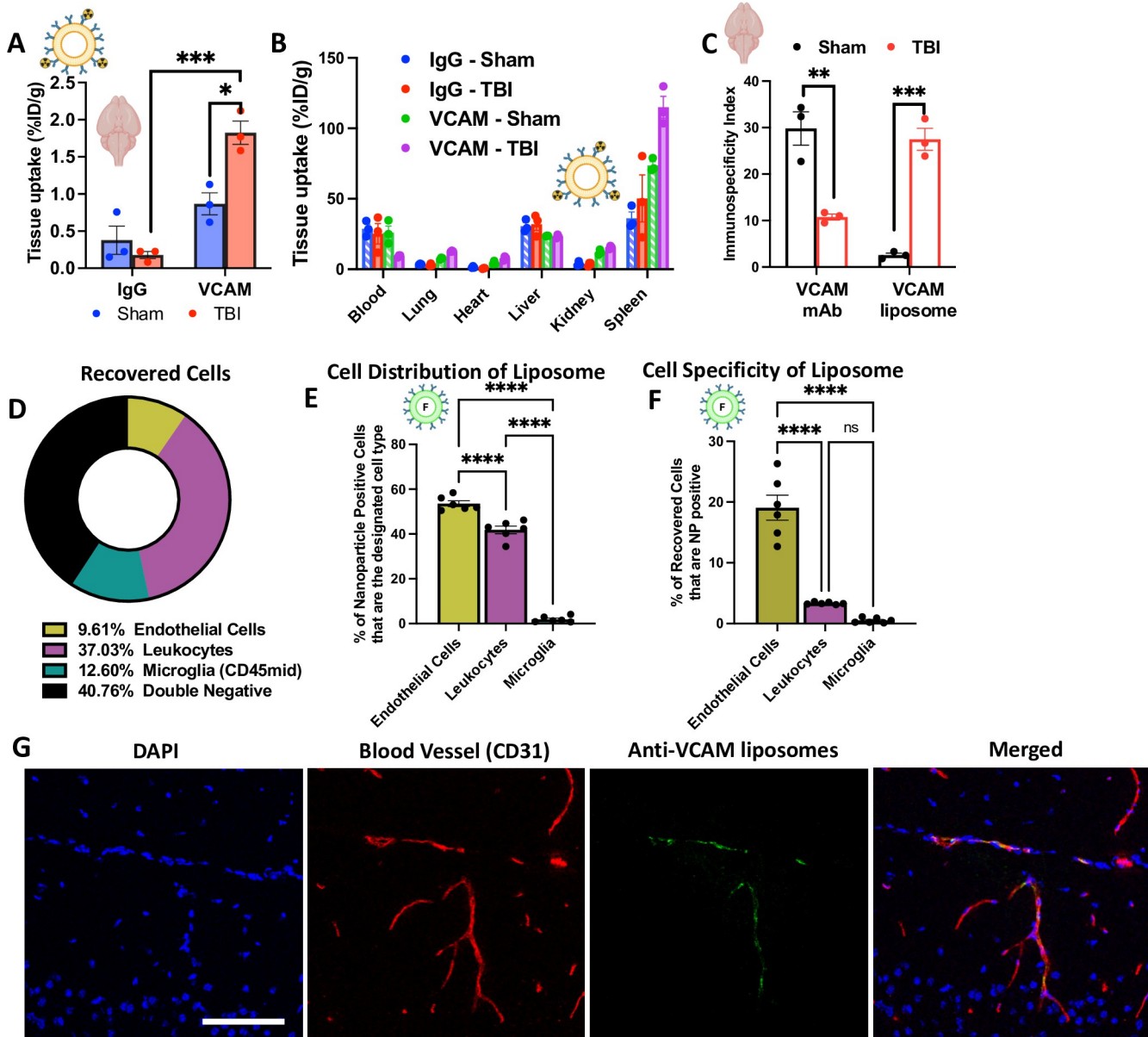

**Fig 2. VCAM-1-targeted liposomes achieve the highest brain uptake in TBI mice and accumulate primarily with endothelial cells.** (A) In sham mice, there is no significant difference in brain uptake between VCAM-1 and IgG liposomes, but in TBI mice, VCAM-1 liposomes are taken up in the brain significantly more than IgG liposomes. (B) No significant differences were observed in the biodistribution of VCAM-1 and IgG liposomes in other organs between sham and TBI mice. (C) Immunospecificity index shows that VCAM-1 antibody has lower targeting specificity in TBI brain. In contract, VCAM-1-targeted liposomes exhibited 10-fold higher targeting specificity to TBI brain, compared to sham. (D) Several cellular populations were recovered during brain flow cytometry: endothelial cells (CD31-high, CD45-neg), leukocytes (CD45-high) and microglia (CD45-mid), and a significantly higher proportion of leukocytes were recovered compared to endothelial cells. Figure calculated as total of 100%. (E) Among all the nanocarrier-positive cells, VCAM-1-liposomes primarily associated with endothelial cells. (F) Within specific cell type, endothelial cells represented a significantly higher proportion of VCAM-1-liposome-positive cells than leukocytes. (G) Using confocal microscopy, we demonstrate the delivery of VCAM-1-targeted liposomes to the endothelium (CD31+) of TBI mouse brains. Scale bar = 100 μm. N≥3 and data shown represents mean ± SEM; For (E), statistical analysis was performed using two-way ANOVA with Sidak's multiple comparisons test. For all other figures, comparisons were made by 1-way ANOVA with Dunnett's post hoc test. *p<0.05, **p<0.01, ***p<0.001, ****p<0.0001.

immunospecificity index. However, the targeting specificity of VCAM-1-liposomes is increased by more than 10-fold following TBI (**Fig 2C**).

Having demonstrated significant brain uptake of VCAM-1-liposomes in TBI mice, we sought to determine their cell type distribution. Fluorescently-labeled VCAM-1 liposomes were injected into mice 24 hours after TBI and brains were harvested and prepared for flow cytometry after 30 minutes of circulation. The following cell-type markers were used: endothelial cells (CD31-high, CD45-neg), leukocytes (CD45-high) and microglia (CD45-mid). As shown in **Fig 2D**, leukocytes took up the highest fraction in the recovered cells (36.35 ± 3.72% for leukocytes vs 9.43 ± 1.08% for endothelial cells and 12.37 ± 1.254 for microglia, also in **S3 Table** **and S1 Fig**). In terms of cellular distribution of VCAM-1-liposomes (**Fig 2E**), out of 100% liposome-positive cells, 53.55 ± 1.255% were endothelial cells, 41.24 ± 1.96% were leukocytes and 1.94 ± 0.67% were microglia. In terms of cell uptake specificity (**Fig 2F**), within the certain cell type, 19.07 ± 2.06% of endothelial cells took up VCAM-1-liposomes compared to only 3.33 ± 0.08% of leukocytes and 0.05 ± 0.17% of microglia. Thus, VCAM-1 liposomes demonstrated preferential endothelial uptake, even in the highly inflamed milieu of TBI, where leukocytes are actively seeking particulate matter to phagocytose.

To further visualize the localization of VCAM-1-liposomes in the post-TBI brain, we injected fluorescently labeled VCAM-1 liposomes into mice 24h after TBI for a 30-minute circulation time. To stain the brain endothelium, fluorescent PECAM/CD31 antibody was injected 5 minutes before sacrifice. Mouse brains were then harvested and sectioned for confocal microscopy. **Fig 2G** shows that VCAM-1-liposomes do indeed localize to the endothelium (CD31+ cells).

## Discussion

Acute brain injuries of various ethiologies display the common pathology of neurovascular inflammation. Upon inflammation, endothelial cells along the BBB are activated and overly express cellular adhesion molecules, including VCAM-1. In contrast to IgG, which is constantly recycled back to the plasma via the neonatal Fc receptor after cell uptake [40], IV administration of VCAM-1 antibody binds endothelium once entering the circulation. The high accessibility of BBB endothelium (surface area of ~20 m$^2$ [41]) and upregulated expression after neurovascular inflammation in TBI makes VCAM-1 a promising target for drug delivery to the brain.

In this study, we found that antibodies and nanocarriers targeted to brain endothelial epitopes accumulate in the brain at higher levels than untargeted IgG controls. In the most powerful example of this, VCAM-1-targeted liposomes achieved more than 10-fold higher delivery to the brain than untargeted IgG-conjugated nanocarriers. Further, we showed that TBI itself does indeed augment uptake of VCAM-1-nanocarriers by ~2-fold. Finally, we showed that the biodistribution of VCAM-1-free antibodies does not reliably predict that of VCAM-1-nanocarriers, as only VCAM-1 conjugated-nanocarriers, but not VCAM-1 antibody, display augmented brain uptake upon TBI. In retrospect, it might be surprising that the mAb and nanocarrier versions of anti-VCAM-1 produce different pharmacokinetics, as the mAb and nanocarriers are different in size, valency, and accessibility to targets. For example, in the context of the disrupted BBB after TBI, IgG at the size of 4~15 nm, should be able to extravasate across disrupted BBB and accumulate in the brain parenchyma, while the nanocarrier with the size of ~145 nm, cannot directly go through the disrupted BBB. Additionally, nanocarriers are easily recognized by phagocytes once entering the blood stream and accumulate in the liver and spleen. Our lab has extensively studied the comparison of antibody vs. nanocarriers in various animal models of acute injuries, and have previously found other pathologies (e.g., acute

lung injury) that dramatically change the pharmacokinetics profiles of antibodies and nano-carriers [21,30,31].

In addition, our flow cytometry data and histology images have demonstrated that the majority of the VCAM-1-targeted liposomes are associated with endothelial cells with 30 minutes post-IV injection. These results align with our primary focus on endothelial targeting because we believe that repairing the leaky BBB is the most immediate task to prevent the brain damage from leakage of toxic plasma across the BBB. However, since VCAM-1 has been reported to be expressed on many other cell types, including astrocytes [42], future studies of nanocarrier pharmacokinetics and cellular distribution will be performed with an extended time courses.

This small initial study showed for the first time with translatable nanocarriers (liposomes) that targeting to the cerebrovascular can greatly increase nanocarrier delivery to the endothelial cells in cerebral vasculature. This study is a continuation of our series of studies of VCAM-1-targeting to the brain in different injury states. We first showed VCAM-1-based targeting to the brain is greatly augmented in a mouse model of acute neurovascular inflammation [21]. More recently, we showed VCAM-1-targeting significantly outperforms untargeted IgG-conjugated nanocarriers in a mouse model of intracerebral hemorrhage (ICH) and ischemic stroke [30,31]. Compliance with the finding of previous studies, this TBI study showed augmented VCAM-1-targeting.

The results of this first study to target the BBB endothelium for drug delivery after TBI warrant further investigation. Specifically, the current data suggest that VCAM-1-targeted liposomes and lipid nanoparticles (LNPs) could contain therapeutics to close the BBB. These therapeutics might include small molecule drugs (e.g., fingolimod) [43] and RNAs (such as mRNA encoding anti-inflammatory proteins), as a strategy to reduce poor long term outcomes following a TBI. Thus, this represents a new strategy to protect the brain from acute and persisting opening of the BBB after TBI.

## Supporting information

**S1 Fig. Recovered cells in TBI brain.** N = 6, mean±SEM.
(DOCX)

**S1 Table. Biodistribution of mAb in sham vs TBI mouse.** N≥3, mean±SEM.
(DOCX)

**S2 Table. Biodistribution of liposomes in sham vs TBI mouse.** N≥3, mean±SEM.
(DOCX)

**S3 Table. Recovered cells in TBI brain.** N = 6, mean±SEM.
(DOCX)

## Acknowledgments

We thank Muzykantov and Brenner lab members for their technical and mental support. We also thank Hui Wei for her help with imaging.

## Author Contributions

**Conceptualization:** Jia Nong, Oscar A. Marcos-Contreras, Jean-Pierre Dollé, Douglas H. Smith, Jacob S. Brenner.

**Data curation:** Serena Omo-Lamai, Jia Nong, Krupa Savalia, Brian J. Kelley, Jichuan Wu, Sahily Esteves-Reyes, Oscar A. Marcos-Contreras.

**Formal analysis:** Serena Omo-Lamai, Jia Nong.

**Funding acquisition:** Vladimir R. Muzykantov, Jacob S. Brenner.

**Investigation:** Jia Nong.

**Methodology:** Serena Omo-Lamai, Jia Nong, Krupa Savalia, Brian J. Kelley, Jichuan Wu, Sahily Esteves-Reyes, Liam S. Chase, Oscar A. Marcos-Contreras.

**Supervision:** Jia Nong, Vladimir R. Muzykantov, Jean-Pierre Dollé, Douglas H. Smith, Jacob S. Brenner.

**Validation:** Jia Nong.

**Visualization:** Jia Nong.

**Writing – original draft:** Serena Omo-Lamai, Jia Nong.

**Writing – review & editing:** Serena Omo-Lamai, Jia Nong, Krupa Savalia, Brian J. Kelley, Vladimir R. Muzykantov, Oscar A. Marcos-Contreras, Jean-Pierre Dollé, Douglas H. Smith, Jacob S. Brenner.

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
