## [Decision Letter · Decision Letter 0]

28 Jul 2023

PONE-D-23-18527Targeting of Nanoparticles to the Cerebral Vasculature After Traumatic Brain InjuryPLOS ONE

Dear Dr. Nong,

Thank you for submitting your manuscript to PLOS ONE. After careful consideration, we feel that it has merit but does not fully meet PLOS ONE’s publication criteria as it currently stands. Therefore, we invite you to submit a revised version of the manuscript that addresses the points raised during the review process. This is important basic research leading to the treatment of traumatic brain injury. The authors are available to respond to the reviewers' comments. Please consider the reviewers' comments and revise your manuscript.

We look forward to receiving your revised manuscript.

Kind regards,

Kazuhiko Kibayashi

Academic Editor

PLOS ONE

Journal Requirements:

2. Please expand the acronym “NIH” (as indicated in your financial disclosure) so that it states the name of your funders in full.

"J.N. received funding from the American Heart Association (Grant 916172). O.A.M.-C. received funding from the American Heart Association (Grant 19CDA34590001). S·O. received funding from the American Heart Association (Grant 23PRE1014444).  J.S.B. received support from the Cardiovascular Institute of the University of Pennsylvania. J.S.B. received funding from K08-HL-138269, R01-HL-153510, R01-HL-160694, R01-HL-157189, R21-AI-166778-01. "

"J.N. received funding from the American Heart Association (Grant 916172). O.A.M.-C. received funding from the American Heart Association (Grant 19CDA34590001). S·O. received funding from the American Heart Association (Grant 23PRE1014444).  J.S.B. received support from the Cardiovascular Institute of the University of Pennsylvania. J.S.B. received funding from K08-HL-138269, R01-HL-153510, R01-HL-160694, R01-HL-157189, R21-AI-166778-01. 

AHA: https://www.heart.org/en/get-involved/ways-to-give?form=FUNPHPZDXBX&s_src=23L511AEMG&s_subsrc=fy23_jun_sem_google_text_&utm_medium=paid&utm_campaign=dr+fy23+june&utm_source=sem+google&utm_content=prospecting-remarketing+sem+general&utm_term=text&gad=1&gclid=CjwKCAjwp6CkBhB_EiwAlQVyxUMZ_FbtTs7VO9Lig6RdFLrTCBLy_fDCRjQoyJToLTNRKw3B3axoQhoC4ZwQAvD_BwE&gclsrc=aw.ds

CVI: https://www.med.upenn.edu/cvi/funded-dream-teams.html

NIH: https://www.nih.gov/grants-funding

The funders had no role in study design, data collection and analysis, decision to publish, or preparation of the manuscript"

Additional Editor Comments:

Dear authors:

This is important basic research leading to the treatment of traumatic brain injury. Basic research on the treatment of traumatic brain injury is few and valuable and the authors are encouraged to promote their research.

Kazuhiko Kibayashi

Reviewers' comments:

Reviewer's Responses to Questions

**Comments to the Author**

1. Is the manuscript technically sound, and do the data support the conclusions?

Reviewer #1: Yes

Reviewer #2: Partly

Reviewer #3: Yes

Reviewer #4: No

2. Has the statistical analysis been performed appropriately and rigorously? 

Reviewer #1: Yes

Reviewer #2: Yes

Reviewer #3: N/A

Reviewer #4: Yes

3. Have the authors made all data underlying the findings in their manuscript fully available?

Reviewer #1: Yes

Reviewer #2: Yes

Reviewer #3: Yes

Reviewer #4: Yes

4. Is the manuscript presented in an intelligible fashion and written in standard English?

Reviewer #1: Yes

Reviewer #2: Yes

Reviewer #3: Yes

Reviewer #4: Yes

5. Review Comments to the Author

Reviewer #1: This manuscript reports on targeting attempts to enable uptake of anti VCAM-antibody targeted liposomes in an experimental model of traumatic brain injury (TBI). This manuscript provides novel observations but revisions are needed.

1) It is said that the blood-brain barrier (BBB) is permeable for years in TBI. Relevant citations are provided, but it would improve this statement if the authors could describe some more on the evidence for this disruption (increases in paracellular or transcellular, persistent inflammation, quantitative measures, compensatory changes in TJ-protein expression patterns etc)

2) The VCAM targeting approach is not all new. Adequate citation is needed (e.g. PMID: 24389338)

3) VCAM: Change to VCAM-1 (There are more VCAM's)

4) The abstract does not read that well: It says testing in TBI increases uptake of VCAM-1-liposomes compared to non-targeted 10-fold. And 2-fold compared to sham (non-TBI ?), but then it says " VCAM antibodies did not demonstrate a similar enhancement in brain uptake following TBI". How does that relate?

5) It is well-known that VCAM-1 liposomes target the brain endothelium, and this may have therapeutic potential, e.g. to enable drug delivery specifically to enhance the endothelium to repair after TBI. A question would be how much is getting further into the brain since the BBB is said to be open for long and whether is could be transformed into something therapeutically useful even when attempted targeted to the brain endothelium (VCAM-1 is probably also expressed by astrocytes). The morphological data of Fig 2 does suggest the liposomes moving further into the brain; please this fact in relation to the proclaimed higher transport after TBI.

Reviewer #2: In this manuscript, authors suggest that VCAM liposomes efficiently target the brain in TBI mice, especially endothelial cells. Their results imply that VCAM liposome may be a novel strategy for drug delivery to the brain and targeting the BBB. However, some points need to be improved for publication in the PLOS ONE.

1) According to Fig.2B, VCAM liposomes are predominantly distributed in the spleen. The brain content of VCAM liposomes may be fairly low. How percentage of VCAM liposomes is achieved in the brain by IV administration? Data is needed to be shown.

2) I wonder why VCAM liposomes had a higher brain uptake than IgG liposomes in the TBI brain. Please describe in the discussion.

3) In the Text, Fig1B, C, D, E, and F were not correctly applied. Please checked carefully.

Reviewer #3: The article by J. Nong et al aims at using a targeted drug delivery approach towards treatment of traumatic brain injury (TBI), which is a quite relevant topic to address. Thus, the authors proposed the use of liposomes decorated with antibodies specific for VCAM (vascular cell adhesion molecule), which is a specific endothelial surface marker, and compared their targeting ability with that of the free antibodies in a mouse model of TBI utilizing central fluid percussion. The main conclusion drawn is that VCAM-liposomes are a promising delivery system for the targeted delivery of therapeutics to the brain following traumatic brain injury. Indeed, VCAM-targeted liposomes have shown a 10-fold higher accumulation in the brain compared to untargeted liposomes, and brain uptake of VCAM-liposomes increased 2-fold in comparison to sham mice. Interestingly, flow cytometry and confocal microscopy analysis demonstrated that VCAM liposomes were primarily taken up by brain endothelial cells post-TBI.

The study presented herein is quite relevant considering the importance of crossing BBB for treating brain diseases. The work is sound and the conclusions are in accordance with the main results obtained. Therefore, the article can be accepted in PLOS after minor revision:

Pages 9/10 - Antibody modification. It seems that there is a missing reference: “ …For biodistribution studies, monoclonal antibodies were radiolabeled with Na125I using Pierce Iodogen radiolabeling method (add reference?). Briefly, tubes…”;

Page 12 - Please replace (Fig 1E) by (Fig 1F);

Page 13 – Discussion. The authors state that “Finally, we showed that the biodistribution of VCAM-free antibodies does not reliably predict that of VCAM-nanocarriers, as only VCAM conjugated-nanocarriers, but not VCAM antibody, display augmented brain uptake upon TBI.” This issue should be discussed into more detail. Is there a tentative reason that explain this behavior? Are there any experimental results of Nong´s Group or literature data that could explain the difference between of VCAM-nanocarriers and free antibodies?

Reviewer #4: The manuscript by Omo-Lamai et al focuses on a vascular cell adhesion molecule (VCAM) -targeting liposome to target inflamed endothelium after TBI. Using a mouse model of TBI, this study shows that intravenous administration of VCAM-targeted liposomes exhibits higher accumulation accumulation in TBI mice, as compared to healthy mice. Flow cytometry and confocal microscopy analysis confirmed that VCAM liposomes were primarily taken up by brain endothelial cells post-TBI. Use of VACM-targeted nanoparticles for different brain applications has been demonstrated previously in multiple papers. Therefore, novelty of this work is limited. The claims are not justified by the data. For example, the authors mention in the abstract that

“intravenous administration of VCAM-targeted nanocarriers (liposomes) exhibited a 10-fold higher accumulation in the brain compared to untargeted liposomes.” But there is no data showing accumulation/uptake of untargeted liposomes. The work is also extremely limited with just looking at brain accumulation, and data has been duplicated in figures. Please see below for a few more specific comments:

1. The introduction needs more clarity regarding what the authors are targeting. For example, is the goal to just target inflamed endothelium in the BBB or cross the BBB. Seems like the focus is targeting endothelium. If that’s true, what is the reason for discussing nanocarriers developed previously for BBB permeation.

2. For data in Fig. 1D and E, authors should show representative images of brain to show antibody accumulation?

3. The authors mention that “VCAM antibody were taken up ~2.5 fold more than IgG. These lower fold improvements in TBI were driven by an increased IgG accumulation in the brain, which is likely due to vascular leak of plasma into brain parenchyma”. This seems more of speculation. Would be great to add imaging data to prove this.

4. The authors say “liposomes, which are FDA approved particles”. Particles are not approved by the FDA, drugs are. It would be better to say “which are present in FDA approved formulations”

5. The authors describe results in Fig. 2A as tissue/brain uptake and also mention

“brain accumulation”, which are not technically correct as their imaging data shows particles binding to endothelium.

6. Data for Vcam mAb in 2C seems like the exact same data in 1D and E. Similarly the data for Vcam liposome in 2C is exactly same as 2A. There is no need for duplication of data.

7. The difference between 2E and F is not clear.

6. PLOS authors have the option to publish the peer review history of their article (what does this mean?). If published, this will include your full peer review and any attached files.

Reviewer #1: No

Reviewer #2: No

Reviewer #3: No

Reviewer #4: No

---

## [Author Response · Author response to Decision Letter 0]

25 Sep 2023

We thank all four reviewers for their constructive feedback and suggestions for our manuscript throughout the entire editorial process. In order to address these comments as thoroughly as possible, we have performed additional experiments and have reanalyzed the existing data. Our point-by-point response to the specific questions raised by the reviewers can be found in the file 'Response to Reviewers'. All changes to the manuscript are denoted with blue text in the file 'Revised Manuscript with Tracked Changes'. In addition, we have updated the format and other information required by the editorial office.

---

## [Decision Letter · Decision Letter 1]

12 Oct 2023

PONE-D-23-18527R1Targeting of Nanoparticles to the Cerebral Vasculature After Traumatic Brain InjuryPLOS ONE

Dear Dr. Nong,

Thank you for submitting your manuscript to PLOS ONE. After careful consideration, we feel that it has merit but does not fully meet PLOS ONE’s publication criteria as it currently stands. Therefore, we invite you to submit a revised version of the manuscript that addresses the points raised during the review process.

 The reviewers and I have reviewed the revised manuscript. One reviewer and I have comments on the revisions, and we would appreciate your response again.

We look forward to receiving your revised manuscript.

Kind regards,

Kazuhiko Kibayashi

Academic Editor

PLOS ONE

Journal Requirements:

Additional Editor Comments:

1. In the TIFF image of Figure 1, the resolution of A, B, and C seems to be low, so please replace them with clearer images if possible.

2. In the figures, "Tissue update (u in lower case)" and "Tissue Update (U in upper case)" are mixed up, so please unify the correct one.

3. Introduction, paragraph3: If the RBC needs to be spelled out, do so. If it is not necessary, there is no need to modify it.

Reviewers' comments:

Reviewer's Responses to Questions

**Comments to the Author**

Reviewer #1: (No Response)

Reviewer #2: (No Response)

Reviewer #3: All comments have been addressed

2. Is the manuscript technically sound, and do the data support the conclusions?

Reviewer #1: Yes

Reviewer #2: Yes

Reviewer #3: Yes

3. Has the statistical analysis been performed appropriately and rigorously? 

Reviewer #1: Yes

Reviewer #2: Yes

Reviewer #3: Yes

4. Have the authors made all data underlying the findings in their manuscript fully available?

Reviewer #1: Yes

Reviewer #2: Yes

Reviewer #3: Yes

5. Is the manuscript presented in an intelligible fashion and written in standard English?

Reviewer #1: Yes

Reviewer #2: Yes

Reviewer #3: Yes

6. Review Comments to the Author

Reviewer #1: I think the authors revised the ms thoroughly and generally replied adequately to the criticism made by the reviewers

Reviewer #2: In this revised manuscript, most points were improved. However, some minor points are needed to be revised for the publication.

1) In the Table S2, value of the brain uptake for VCAM liposome (sham) may be not consistent with that of Figure 2A.

Please check carefully.

2) Figure Legend: In the Fig 1(F), “PECAM” is described although data is not shown in the text and Figures. Please

delete if data of PECAM is not shown.

3) In the Fig 2C, I wonder why targeting specificity of VCAM mAb was deceased by TBI.

Reviewer #3: (No Response)

7. PLOS authors have the option to publish the peer review history of their article (what does this mean?). If published, this will include your full peer review and any attached files.

Reviewer #1: No

Reviewer #2: No

Reviewer #3: No

---

## [Author Response · Author response to Decision Letter 1]

16 Nov 2023

The response to reviewers has been uploaded in a separate file. The additional comments from the editor have been addressed in the revised manuscript.

---

## [Editor Report · Decision Letter 2]

7 Dec 2023

PONE-D-23-18527R2Targeting of Nanoparticles to the Cerebral Vasculature After Traumatic Brain InjuryPLOS ONE

Dear Dr. Nong,

Thank you for submitting your manuscript to PLOS ONE. After careful consideration, we feel that it has merit but does not fully meet PLOS ONE’s publication criteria as it currently stands. Therefore, we invite you to submit a revised version of the manuscript that addresses the points raised during the review process.

Dear aothors:Please consider the following comments from the editor.

We look forward to receiving your revised manuscript.

Kind regards,

Kazuhiko Kibayashi

Academic Editor

PLOS ONE

Journal Requirements:

Additional Editor Comments:

Dear aothors:

I have a request regarding the reproduction of the images in Figure 1 from another article. You state that the images are reproduced with permission [39, 40].

You should indicate which images are taken from existing articles. Also indicate that you have permission to reproduce the images from both the authors and the publishers of the articles from which the images are taken.

Please obtain written permission from both the authors and the publishers of the articles from which the images are taken. Please include the written permissions with your submission.

Kazuhiko Kibaayshi

Academic Editor

---

## [Author Response · Author response to Decision Letter 2]

2 Jan 2024

We thank the editor for the feedback and guidance regarding the reproduction of images in Figure 1. We appreciate your thorough review, and we have revised the manuscript and confirmed that all the images in the updated Figure 1 are original and not taken from existing articles. We have removed the incorrect statement in the caption of Figure 1. In addition, we have changed the figures with individual dots to present all the raw data included in this manuscript in both main figures and supporting document.

---

## [Editor Report · Decision Letter 3]

4 Jan 2024

Targeting of Nanoparticles to the Cerebral Vasculature After Traumatic Brain Injury

PONE-D-23-18527R3

Dear Dr. Nong,

We’re pleased to inform you that your manuscript has been judged scientifically suitable for publication and will be formally accepted for publication once it meets all outstanding technical requirements.

Kind regards,

Kazuhiko Kibayashi

Academic Editor

PLOS ONE

Additional Editor Comments:

Dear authors:

I understand that all images in Figure 1 are original. I have no request for further revision.

Since there is no effective treatment for traumatic brain injury, basic research on treatment methods is extremely important. I sincerely hope that the authors will continue to develop basic research that will lead to the treatment of traumatic brain injury.

Kazuhiko Kibayashi
---

## [Editor Report · Acceptance letter]

17 May 2024

PONE-D-23-18527R3 

PLOS ONE

Dear Dr. Nong, 

I'm pleased to inform you that your manuscript has been deemed suitable for publication in PLOS ONE. Congratulations! Your manuscript is now being handed over to our production team.

Kind regards, 

on behalf of

Professor Kazuhiko Kibayashi 

Academic Editor

PLOS ONE